# Efficacy and Safety of Adding Ribavirin to Sofosbuvir-Based Direct-Acting Antivirals (DAAs) in Re-Treating Non-Genotype 1 Hepatitis C—A Systematic Review and Meta-Analysis

**DOI:** 10.3390/diseases13050138

**Published:** 2025-04-29

**Authors:** Shahd Hamran, Amani A. Al-Rajhi, Kawthar Jasim, Majed A. Al-Theyab, Mohamed Elahtam, Mooza K. Al-Hail, Wadha Al-Fahaidi, Yaman A. Khamis, Yara Dweidri, Abdel-Naser Elzouki, Tawanda Chivese

**Affiliations:** 1College of Medicine, QU Health, Qatar University, Doha P.O. Box 2713, Qatar; sh2004556@qu.edu.qa (S.H.); aa1802610@student.qu.edu.qa (A.A.A.-R.); ma2001971@student.qu.edu.qa (M.A.A.-T.); me1901983@student.qu.edu.qa (M.E.); ma1901280@student.qu.edu.qa (M.K.A.-H.); wa1905149@student.qu.edu.qa (W.A.-F.); yk1907250@student.qu.edu.qa (Y.A.K.); yd1907469@student.qu.edu.qa (Y.D.); 2Internal Medicine, Hamad Medical Corporation, Doha P.O. Box 3050, Qatar; aelzouki@hamad.qa; 3College of Dental Medicine, QU Health, Qatar University, Doha P.O. Box 2713, Qatar; kawthar.jasim@qu.edu.qa; 4Division of Science and Mathematics, School of Interdisciplinary Arts and Sciences, University of Washington Tacoma, Washington, WA 98402, USA

**Keywords:** hepatitis C virus (HCV), ribavirin (RBV), direct-acting Antivirals (DAAs), HCV genotype, sustained virological response (SVR), systematic review and meta-analysis

## Abstract

Background: There is still debate whether ribavirin should be added to direct-acting antivirals (DAAs) for the management of treatment-experienced individuals with non-genotype-1 hepatitis C. This study compared the efficacy and safety of adding ribavirin to sofosbuvir-based combinations compared to sofosbuvir-based regimens alone in treating non-genotype 1 hepatitis C virus (HCV) in individuals who have been previously treated. Methods: We searched Cochrane CENTRAL, PubMed, SCOPUS, CINAHL and preprint databases from inception to September 2023 for randomized controlled trials (RCTs) that compared sofosbuvir-based regimens with ribavirin to sofosbuvir-based regimens alone in previously treated individuals with non-genotype 1 HCV infection. Data extraction and quality of study assessments were performed by two independent authors, and synthesis was performed using bias-adjusted models, heterogeneity using I2, and publication bias using funnel plots. Results: Eight RCTs compared sofosbuvir-based combinations with and without ribavirin were included. Overall, the addition of ribavirin to sofosbuvir, compared to sofosbuvir alone, did not show a benefit in achieving sustained virological response (SVR) (OR 0.91, 95% CI 0.26–3.17, I2 = 70.0%) with moderate certainty in Grading of Recommendations, Assessment, Development, and Evaluation (GRADE) evidence. In subgroup analysis, there was no benefit of adding ribavirin to sofosbuvir in individuals with non-genotype 1 HCV. The additional ribavirin was associated with increased adverse events (OR 2.03, 95% CI 1.58–2.6, I2 = 8.0%) and treatment discontinuation (OR 1.81, 95% CI 0.78–4.28, I2 = 0.0%). Conclusions: The moderate certainty evidence suggests that adding ribavirin to sofosbuvir-based regimens may not confer benefit in achieving SVR in previously treated individuals with non-genotype 1 HCV but increases the odds of adverse events and treatment discontinuation. More evidence is needed on the effect of additional ribavirin in achieving SVR in individuals with decompensated cirrhosis. Registration: The protocol is registered on the International Prospective Register of Systematic Reviews (PROSPERO) (CRD42022368868).

## 1. Introduction

Hepatitis C virus (HCV) is a global epidemic that affects an estimated 58 million infected individuals worldwide, resulting in roughly 400,000 deaths each year, primarily from complications including liver cirrhosis and hepatocellular carcinoma [1]. Treatment rates for HCV remain low, with only 13% of people receiving treatment [1]. In the absence of a vaccine, the elimination of this virus will have to be achieved using behavior-based methods to stop transmission and cure existing cases using antiviral therapy. One of the most significant advances in HCV treatment has been the development of direct-acting antivirals (DAAs), which have high cure rates, with up to 95% of treated individuals achieving sustained virological response (SVR) within 12 weeks after treatments [1,2]. DAAs are classified into four classes based on their therapeutic target and mechanism of action [3], and major guideline bodies recommend first-line DAAs that are pan-genotypic (i.e., treat all HCV genotypes), such as sofosbuvir and velpatasvir [4,5,6]. Apart from their potency, DAAs have relatively few side effects when given orally over a short time [1].

Despite the availability of highly efficacious curative DAAs, a significant testing and treatment gap remains. In addition, there are still individuals who require treatment after either failure of DAA therapy or failure of interferon-based regimens, which were previously recommended [7]. Some of the treatment failures have been attributed to certain HCV genotypes, which do not respond well to treatment. For example, genotype 4 did not respond well to interferon-based therapies, while genotype 3 may not respond well to some DAAs [8,9]. HCV has eight common genotypes (genotypes 1 to 8), with genotype 1 being the most common globally and also the most researched, and the genotypes have varying distribution in different geographical regions [5,10,11]. Despite the availability of pangenotypic DAAs, there is still a need for research on optimal treatments for other types of HCV that are not genotype 1 [12].

There is still no consensus on the treatment options for retreating HCV [11]. Current retreatment options usually involve combining sofosbuvir with another class of DAA, extending the treatment duration [11,13], and some guidelines [4,13] recommend the addition of ribavirin to DAAs. Ribavirin, a guanosine analog, has been used as an adjunct to interferon-based therapies, and several trials have investigated its use as an add-on to DAAs in retreating HCV [14,15,16]. Ribavirin acts by directly inhibiting HCV viral replication and HCV RNA polymerase [14]. However, ribavirin is potentially teratogenic, may result in hemolytic anemia decreased lymphocyte counts, and has potential for possible carcinogenicity [17]. Because of its teratogenicity, the drug is contraindicated in women planning to be pregnant [18].

Randomized controlled trials (RCTs) of adding ribavirin to DAAs have produced conflicting results, especially in treatment-experienced participants. For example, two RCTs concluded that adding ribavirin to DAAs did not improve treatment outcomes and increased the risk of side effects in people who required retreatment for HCV [15,19]. However, another RCT demonstrated the benefit of achieving SVR when ribavirin was added to DAAs in retreating HCV [20]. Existing meta-analyses of RCTs have shown no benefit in adding ribavirin to DAAs when treating either treatment-naïve [16] or treatment-experienced participants with genotype 1 HCV [21,22]. However, an evidence gap remains regarding whether ribavirin has benefit when added to DAAs for the retreatment of non-genotype 1 HCV. Therefore, we conducted a systematic review and meta-analysis to assess the efficacy and safety of adding ribavirin to DAAs, particularly to sofosbuvir-based regimens, compared to DAAs alone in the retreatment of individuals with non-genotype 1 HCV.

## 2. Methods

### 2.1. Study Design

This research is a systematic review and meta-analysis and follows the Preferred Reporting Items for Systematic Reviews and Meta-Analyses (PRISMA) guidelines [23] (Appendix A). The protocol for this study is registered in the International Prospective Register of Systematic Reviews (PROSPERO) (CRD42022368868).

### 2.2. Data Sources

We searched the Cochrane Central Register of Controlled Trials (CENTRAL), PubMed, Scopus, Cumulated Index to Nursing and Allied Health Literature (CINAHL), and the databases of preprints such as medRXIV for studies published from January 2010 to November 2022, with no language restrictions. The search was conducted during November 2022, using a search strategy (Appendix A) and updated during September 2023.

### 2.3. Search Methods

#### 2.3.1. Search Terms for DAAs

“sofosbuvir” OR “sovaldi” OR “simeprevir” OR “olysio” OR “daclatasvir” OR “daklinza” OR “ledipasvir” OR “harvoni” OR “elbasvir” OR “grazoprevir” OR “zepatier” OR “velpatasvir” OR “epclusa” OR “ombitasvir” OR “paritaprevir” OR “dasabuvir” OR “viekira pak” OR voxilaprevir OR ritonavir OR 3D OR glecaprevir OR pibrentasvir OR mavyret OR “direct-acting agents” OR “direct acting antiviral” OR daa.

#### 2.3.2. Search Terms for Hepatitis C

“Hepatitis c” OR HCV OR “chronic hepatitis C” OR “Acute Hepatitis C”.

#### 2.3.3. Search Terms for Ribavirin

Ribavirin OR Rebetol OR Ribasphere OR RibaPak OR Copegus OR Virazole OR Moderiba OR “Tribavirin” OR “Vilona” “Viramide” OR “Virazide” OR “ICN-1229” OR “ICN 1229” OR “ICN1229” OR “Ribamide” OR “Ribamidil” OR “Ribamidyl” OR “RBV”.

#### 2.3.4. Search Terms for Hepatitis C Non-Genotype 1

“Genotype 2” OR “genotype 3” OR “genotype 4” OR “genotype 5” OR “genotype 6” OR “GT2” OR “GT3” OR “GT4” OR “GT5” OR “GT6” OR “non-genotype 1” OR “non genotype 1” OR “non GT1”.

### 2.4. Procedure for Selection of Studies

The study records from the search were imported into EndNote for deduplication and, subsequently, uploaded to the Rayyan systematic review management website (https://www.rayyan.ai/) for screening using the title and abstracts. The full text of preliminarily included study records was retrieved and evaluated for eligibility by two independent investigators. In cases of disagreement, a third investigator was consulted to make the final decision.

### 2.5. Eligibility

Studies included were limited to original RCTs that compared the efficacy and safety of Sofosbuvir-based combinations with and without ribavirin in treating non-genotype 1 HCV in treatment-experienced participants [24]. Included studies should have specified the HCV genotype, reported the analyses of treatment-experienced individuals separately, used sofosbuvir across both arms and had full text available. We excluded observational studies, reviews, RCTs with only genotype 1 or treatment-naïve individuals, animal or lab studies, and RCTs that did not report the main outcome of SVR12. In studies with both genotype 1 and non-genotype 1, we extracted data only from non-genotype 1 participants. Moreover, data were extracted for groups of equal treatment duration.

### 2.6. Key Definitions

Retreatment is initiated in patients with treatment failure, defined as the failure to achieve SVR at the end of treatment, which can be due to non-response relapse or re-infection [25].

### 2.7. Outcomes

The primary efficacy outcome was SVR, which is defined as HCV-RNA levels below 15 IU/mL, measured 12 weeks after the end of treatment [26]. The primary safety outcome was treatment discontinuation. The secondary efficacy outcome was SVR at 24 weeks (SVR24) after treatment completion [26]. The secondary safety endpoints were developing any common adverse events, including fatigue, headache, dermatologic manifestations, and gastroenterological symptoms such as nausea, vomiting, and diarrhea [27], as well as serious adverse events.

### 2.8. Data Extraction

Data extraction was performed independently by two authors using Microsoft Excel. The extracted data included data on study design, date, location, sample size, HCV genotype, type of prior treatment, treatment regimen, the number of individuals with SVR at 12 and 24 weeks, side effects, serious side effects, and treatment discontinuation.

### 2.9. Assessing the Quality of Included Studies

The quality of the included studies was assessed using the Methodological Standard for Epidemiological Research (MASTER) scale, comprising seven standards subdivided into 36 safeguards [28].

### 2.10. Data Synthesis

#### Synthesis for Efficacy and Safety of DAAs With and Without Ribavirin

In this study, we recalculated the unadjusted odds ratio (OR) for each study to calculate pooled odds ratios (ORs) for SVR12 and other outcomes, along with their 95% confidence intervals (CIs), using the quality-effects model [29]. The quality-effects model uses random error variance in addition to variance due to systematic error as weights. The systematic error variance weights were derived from quality ranks, which were derived from the quality assessment, thereby adjusting for possible bias in the analysis phase [29]. The random-effects model assumes that the treatment effects obtained from an assumed superpopulation of studies (which do not exist) follow a normal distribution, thus giving spuriously precise Cls. Moreover, because of the increase in the mean squared error (MSE), the point estimate from the meta-analysis has a possibility of deviating away from the true value much more than with the quality-effects model [30,31]. Forest plots were used to display the pooled OR estimates and their CIs. Sensitivity analysis was conducted to re-analyze data without bias adjustment using the quality-effects model and the random-effects model [32]. We used Stata version 17 (College Station, TX, USA) with the metan package for the meta-analysis and reported exact *p* values. We used the I2 statistic, Cochran’s Q *p*-value, and the Galbraith plot to assess heterogeneity and I2 values of 25%, 50%, and 75% were interpreted as low, moderate, and high inconsistency categories, respectively [33]. We conducted subgroup analysis for studies with previous treatment regimens (either DAAs or interferon-based regimens) and for the cirrhosis status. We investigated publication bias using Doi plots and the Luis Furuya-Kanamori (LFK) index [34], in addition to funnel plots and Egger regression [35]. The Grading of Recommendations, Assessment, Development, and Evaluations (GRADE) framework was used to rate evidence quality and recommendation strength for the primary outcome of SVR12 [36].

### 2.11. Ethics

This systematic review utilized published data. Therefore, there was no need for ethical approval.

## 3. Results

### 3.1. Search Results

A total of 7206 studies were identified from all the searches. After the removal of duplicates and manual screening by title and abstract, 112 studies remained for full-text screening, of which 102 were excluded for reasons shown in Figure 1. Most studies excluded after full-text screening the full text were not RCTs (*n* = 53) or did not compare the drugs of interest (*n* = 23). Eight studies were chosen according to the inclusion criteria [19,37,38,39,40,41,42,43]. Two RCTs were excluded because they did not use sofosbuvir in DAAs combination [20,44].

### 3.2. Characteristics of Included Studies

The eight RCTs included [19,37,38,39,40,41,42,43] had a total of 772 participants from eight countries, with three studies from the United States of America, two from Egypt, and the remainder from multiple countries (Table 1). One study focused only on genotype 2 [37], three studies on genotype 3 [39,40,41], two studies on genotype 4 [38,43] and two studies included several genotypes [19,42]. All studies included some participants with compensated cirrhosis, and three of them included individuals with cirrhosis only [19,39,41]. Most studies excluded human immunodeficiency virus (HIV) and hepatitis B virus (HBV) co-infections [19,37,38,40,42,43], but two studies included HIV co-infected participants [39,41]. All studies included participants previously treated with interferons, while three studies also included individuals with prior DAA treatment [19,39,40] (Table 1).

### 3.3. Assessment of the Quality of Included Studies

Most of the included studies achieved MASTER scale scores ranging from 29 to 31 out of 36, indicating an overall high quality of evidence (Appendix A). One study did not meet the first safeguard as they excluded a participant after randomization [38]. Additionally, two studies did not adequately report how allocation concealment was conducted [37,39].

### 3.4. Efficacy of Adding Ribavirin to DAAs in Achieving SVR at 12 Weeks—GRADE Assessment

All eight included RCTs [19,37,38,39,40,41,42,43] with a total of 772 participants, of which 420 were on additional ribavirin, and 352 were on sofosbuvir-based regimens alone, reported data on SVR12. The effect of adding ribavirin to sofosbuvir-based regimens varied slightly across studies; one study showed a reduction in efficacy [40], six showed no effect [19,37,38,39,41,43], and one study reported an increase in efficacy [42]. The studies included were RCTs, which provide high-quality evidence, and therefore, the initial GRADE was set as high certainty, pending assessments of other GRADE domains.

### 3.5. Overall Effect Size and Consistency

In the overall synthesis, the addition of ribavirin to sofosbuvir-based regimens did not improve SVR achievement at 12 weeks (OR 0.91, 95%CI 0.26–3.17), with moderate statistical heterogeneity (I2 = 70.0%) (Figure 2). This effect was the same even when the random-effects model was used (Figure 2). Visual inspection of the forest plot showed some inconsistency, and measures of heterogeneity all suggested some statistical heterogeneity for this outcome (I2 = 70.0%, Cochran’s Q *p* value = 0.001, tau2 = 1.498). The Galbraith plot (Appendix A) showed an effect near the null, represented by an almost horizontal line, and some heterogeneity shown by two studies outside the upper and lower CI. Therefore, the GRADE was downgraded by one level because of the lack of consistency.

### 3.6. Directness, Study Quality, and Publication Bias

In terms of directness, the interventions specified for these participants and the outcome of SVR12 were similar to those of interest to the healthcare system and clinicians. Most of the RCTs had a high-quality rating, and the funnel and Doi plots (Appendix A) suggested that there was no publication bias. Therefore, after the assessment of these domains, the GRADE rating remained as moderate certainty evidence for the primary outcome.

### 3.7. Final GRADE Rating for the Efficacy of Adding Ribavirin to DAAs to Achieve SVR12

The GRADE was downgraded by one level due to heterogeneity in the RCTs, and therefore, the GRADE rating for the evidence for the main outcome was of moderate certainty that adding ribavirin to sofosbuvir-based regimens does not improve efficacy in achieving SVR12.

### 3.8. Sensitivity Analysis

Similar results were observed in the leave-one-out sensitivity analysis (Appendix A).

### 3.9. Subgroup Analysis

When the analysis was stratified by the genotypes, no additional benefit was observed when ribavirin was added for genotype 3 (OR 0.78, 95% CI 0.09–6.63, I2 = 83.2%) and genotype 4 (OR 1.39, 95% CI 0.31–6.31, I2 = 4.7%) (Appendix A). Only one study [37] provided data on participants with genotype 2, and in that study, the addition of ribavirin greatly reduced the efficacy observed (OR 0.36, 95% CI 0.07–2.01). Although subgroup analysis by cirrhosis status due to lack of subgrouped data within the studies, in one study of individuals with decompensated cirrhosis, ribavirin showed benefit when added to sofosbuvir-based regimens, with 85% SVR12 achievement rate with ribavirin compared to 50% SVR12 in individuals without ribavirin (OR 3.97, 95% CI 0.71–22.11) [19].

### 3.10. SVR at 24 Weeks

Only one RCT [38] assessed SVR24. The study suggested no benefit in achieving SVR24 when ribavirin was added to sofosbuvir-based regimens (OR 0.90, 95% CI 0.17–4.67).

### 3.11. Safety of Adding Ribavirin to DAAs

Any adverse events, serious adverse events and discontinuation of treatment.

Data from all eight studies showed increase in the odds of adverse events in the additional ribavirin group, with little to no heterogeneity (OR 2.03, 95% CI 1.58–2.6, I2 = 8.0%) (Figure 3 and Appendix A) and higher odds of discontinuation of treatment (OR 1.81, 95% CI 0.78–4.28, I 2 = 0.0%) (Appendix A). The synthesis also suggested no difference in the odds of serious adverse events (OR 1.13, 95% CI 0.61–2.06, I2 = 21.9%, *n* = 7 studies) (Appendix A).

This forest plot shows the overall analysis of the odds of adverse events in the treatment group (DAAs and ribavirin) compared to the control group (DAAs only), which was reported by all eight studies. The meta-analytic effect size using the Quality effects and Random effects models are (OR 2.03) and (OR 2.03), respectively.

## 4. Discussion

In this meta-analysis of eight RCTs, we found that adding ribavirin to sofosbuvir-based regimens did not improve efficacy in achieving SVR12 in adults previously treated for non-genotype 1 HCV. Additionally, the findings confirm that adding ribavirin to sofosbuvir increases the odds of adverse events and treatment discontinuation.

Our synthesis shows that there is no difference in efficacy when ribavirin is added to sofosbuvir-based regimens, compared to sofosbuvir-based regimens alone with an overall odds ratio of 0.91 of achieving SVR12, with moderate certainty GRADE evidence. These findings are similar to those of other meta-analyses that found no benefit in achieving SVR12 with the addition of ribavirin to DAA regimens in other sub-populations [2,45,46,47,48,49]. For example, in HCV genotype 1, regardless of prior treatment history, two meta-analyses have shown that adding ribavirin to DAAs did not improve the achievement of SVR, with overall risk ratios of one in both analyses [46,48]. Additionally, we found that ribavirin also did not add to the efficacy of sofosbuvir-based regimens when SVR was assessed at 24 weeks, although this result came from one study. Notably, some guidelines [4,50] still recommend the use of ribavirin in retreating HCV, perhaps due to a lack of up-to-date syntheses in non-genotype 1 treatment-experienced populations [11], which this study now provides.

While we restricted our analyses to RCTs that had HCV genotypes determined in each participant, our findings, together with those of existing meta-analyses [45,46,47,48,49,51], suggest that given the availability of pangenotypic DAAs such as sofosbuvir and velpatasvir [11] there may be no need to genotype HCV infections before starting treatment [51]. A recent RCT, which we excluded because of a lack of genotype reporting, also found no benefit in adding ribavirin to DAAs when retreating HCV in Egypt [15]. Notably, the findings from our included studies showed a high degree of efficacy, with treatment success of around 90% in both ribavirin and non-ribavirin trial arms, again confirming the high efficacy of DAAs.

The current meta-analysis confirmed that ribavirin was associated with an increase in the odds of individuals discontinuing treatment. Previous meta-analyses [2,45,46,47,48,49,52] have not reported on treatment discontinuation, which is a key factor influencing outcomes, especially in individuals who need retreatment. However, other meta-analyses have found ribavirin to be associated with a higher risk of adverse events [2,47,52], which aligns with our findings. Notably, ribavirin cannot be used in both male and female individuals who are planning to have children because of its high teratogenicity [18].

The strengths of this study include the inclusion of studies where genotyping was performed, clear eligibility criteria, a comprehensive search of the literature and rigorous analysis, and the use of GRADE evidence certainty, making the results of this study easier to translate to clinical practice and practice guidelines. A limitation of this study is the moderate heterogeneity observed in the synthesis of the primary outcome (SVR12), which we investigated using subgroup analysis. Another limitation is that we did not investigate the appropriate DAA combinations for retreating HCV.

## 5. Conclusions

The moderate certainty evidence suggests that adding ribavirin to sofosbuvir-based regimens may not confer benefit in achieving SVR in previously treated individuals with non-genotype 1 HCV but increases the odds of adverse events and treatment discontinuation. More evidence is needed on the effect of additional ribavirin in achieving SVR in individuals with decompensated cirrhosis.

## Figures and Tables

**Figure 1 diseases-13-00138-f001:**
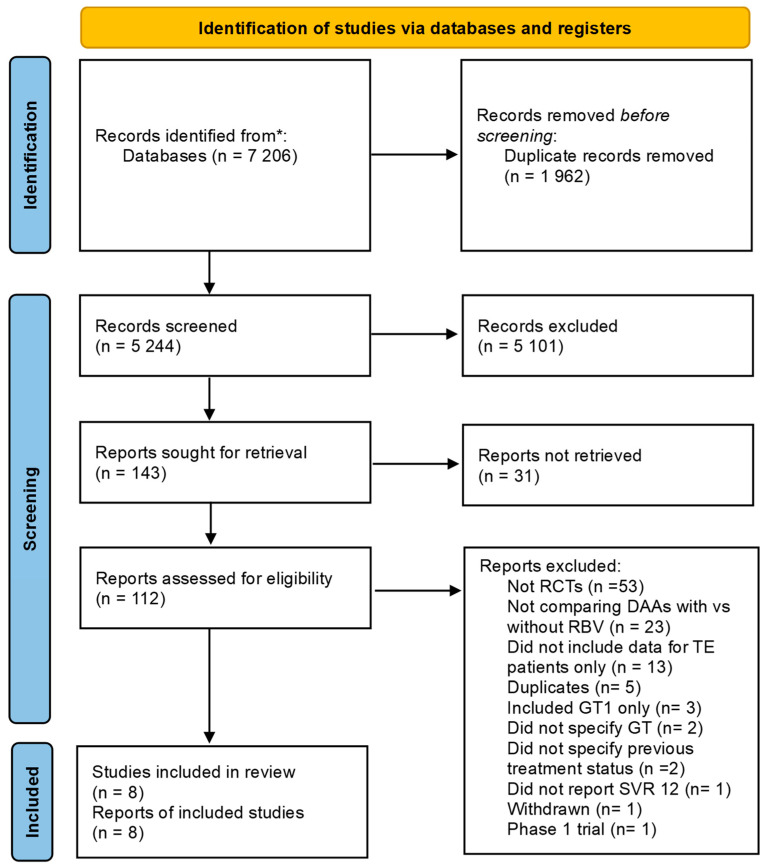
PRISMA flow chart illustrates the screening, exclusion, and selection of the studies in the meta-analysis. Abbreviations: (*) Identified from multiple sources (CENTRAL, PubMed, Scopus, CINAHL, and the databases of preprints such as medRXIV), Randomized controlled trials (RCTs), Direct-acting Antiviral (DAA), Ribavirin (RBV), Treatment Experienced (TE), Genotype (GT), Sustained Viral Response (SVR).

**Figure 2 diseases-13-00138-f002:**
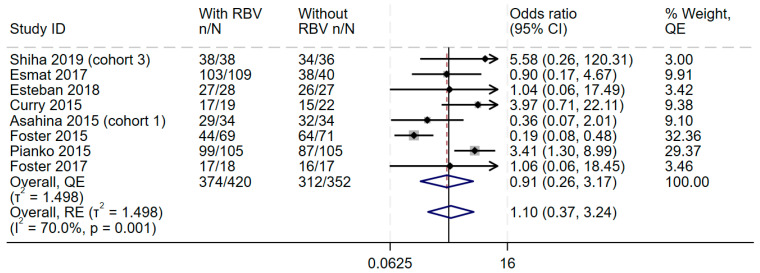
Efficacy of adding ribavirin to sofosbuvir-based regimens compared to sofosbuvir-based regimens alone. The vertical dashed line at OR = 1 represents the line of no effect. The diamond at the bottom represents the pooled overall estimate from all studies, with its center indicating the overall OR and its width denoting the 95% CI [19,37,38,39,40,41,42,43]. Abbreviations: Confidence interval (CI), Direct-acting Antiviral (DAA), Genotype (GT), interferon (IFN), Quality Effects (QE), Random Effects (RE).

**Figure 3 diseases-13-00138-f003:**
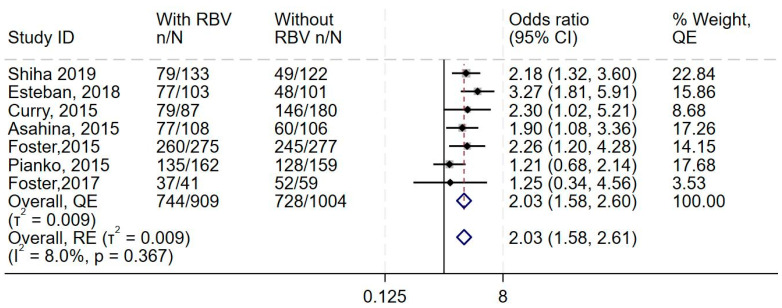
Effect of adding ribavirin to sofosbuvir-based regimens on adverse events. The vertical dashed line at OR = 1 represents the line of no effect. The diamond at the bottom represents the pooled overall estimate from all studies, with its center indicating the overall OR and its width denoting the 95% CI [19,37,39,40,41,42,43].

**Table 1 diseases-13-00138-t001:** Characteristics of included studies.

Author/ Publication Year	Country	Follow-Up Period (Months)	Type of Prior Treatment	GT Included	Sample Size	Inv Group Regimen + Duration	Control Group Regimen + Duration	Cirrhosis
Shiha 2019 (cohort 3) [43]	Egypt	3	IFN	GT4	74	LDV/SOF + RBV for 12 or 16 weeks	LDV/SOF for 12 weeks	28% int, 26% in control
Esmat 2017 [38]	Egypt	6	IFN + RBV	GT4	149	RDV/SOF + RBV for 12 weeks	RDV/SOF for 12 weeks	47% in total TE
Esteban, 2018 [39]	Spain	3	DAA +PEG-IFN + RBV/PEG-IFN+ RBV	GT3	55	SOF/VEL + RBV for 12 weeks	SOF/VEL for 12 weeks	100%
Curry, 2015 [19]	USA	6	DAA/PEG-IFN + RBV	GT1 (1a, 1b), GT2, GT3, GT4	41	SOF/VEL + RBV for 12 weeks	SOF/VEL for 12 or 24 weeks	100% (decompensated)
Asahina, 2015 [37]	Japan	6	DAA/PEG-IFN	GT2 (2a, 2b, and 2c)	68	SOF + RBV for 12 weeks	LDV/SOF for 12 weeks	13% int, 15% control (for TN and TE)
Foster, 2015 [40]	USA, Canada, Europe, Australia and New Zealand	6	DAA + PEG-IFN + RBV and PEG-IFN + RBV and others	GT3	142	SOF + RBV for 24 weeks	SOF/VEL for 12 weeks	52% in int, 55% in control
Pianko, 2015 [42]	Australia, New Zealand, USA	3	IFN	GT3, GT1	210	SOF/VEL + RBV for 12 weeks	SOF/VEL for 12 weeks	50% in int, 49% in control
Foster, 2017 [41]	UK	6	PEG-IFN/RBV	GT3	35	SOF/EBR/GZR + RBV for 12 weeks	SOF/EBR/GZR for 12 or 16 weeks	100%

Abbreviations: Direct-acting Antiviral (DAA), Elbasvir (EBR), Genotype (GT1), Genotype 2 (GT2), Genotype 3 (GT3), Genotype 4 (GT4), Grazoprevir (GZR), Interferon (IFN), Intervention (int), Ledipasvir (LDV), Pegylated Interferon (PEG—IFN), Remdesivir (RDV), Ribavirin (RBV), Sustained Viral Response (SVR), Sofosbuvir (SOF), Treatment Experienced (TE), Treatment Naive (TN), United Kingdom (UK), United States of America (USA), Velpatasvir (VEL).

## Data Availability

The original data presented in the study are openly available in the published RCTs.

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
