# Peer review of "Efficacy and Safety of Adding Ribavirin to Sofosbuvir-Based Direct-Acting Antivirals (DAAs) in Re-Treating Non-Genotype 1 Hepatitis C—A Systematic Review and Meta-Analysis"

_diseases, 2025, doi:10.3390/diseases13050138_

Round 1

Reviewer 1 Report

Comments and Suggestions for Authors

How about the efficacy of DAA without ribavirin in cirrhotic patient? (SVR rate no difference?)

Comments on the Quality of English Language

No further comment 

Author Response

Response to reviewers

In reference to:

Manuscript ID: diseases-3498962

Type of manuscript: Systematic Review

Title: Efficacy and safety of adding ribavirin to sofosbuvir-based direct-acting antivirals

(DAAs) in re-treating non-genotype1 hepatitis C- a systematic review and

meta-analysis.

Reviewer number 1 comment:

  • How about the efficacy of DAA without ribavirin in cirrhotic patients? (SVR rate no difference?)

Thank you for your comment. In our analysis of eight RCTs we could not compare SVR outcome in patients with vs without cirrhosis, as the included primary studies did not have this direct comparison. However, most but not all patients in the included studies had cirrhosis, meaning that our findings could potentially be applied to this population.

Reviewer number 2 comment:

  • The references could be updated with more recent ones.

Thank you for your comment. The references have been updated.

Reviewer number 3 comments:

A few suggestions for improvement follow:

  1. Change the title to make clear that non-sofosbuvir based treatments were excluded from the metanalysis ie the conclusion is only regarding adding ribavirin to sofosbuvir-based regimens in re-treatment of non-G1 patients.

Thank you for your comment, we agree with the suggestion and have applied it accordingly.

  1. The paper reads well, but the abstract is a little stilted and the English could be improved. eg line 28 change "different HCV genotypes" to "any non-1 HCV genotype" line 29 "The additional ribavirin showed increased odds of developing adverse events" change to "The addition of ribavirin was associated with increased adverse effects"

Thank you for your suggestions. We have now made these changes.

  1. Overall I am happy that the paper is ready for publication with only minor revisions.

Thank you so much for providing us with feedback.

Reviewer number 4 comments:

Major concerns:

  • Since it directly affects the findings of these studies, the review must focus on the characteristics of the specific patient populations analyzed in the eight clinical trials presented, in addition to the analysis's methodology and validity.
    Many thanks for your informative remarks on our manuscript. We appreciate being given the chance to improve our work in accordance with your input. We have a separate section in our paper that describes the patient populations studied across all of the eight included studies. We present a table that summarizes the characteristics of the studies. In addition, we have updated our discussion to reflect the possible impact of the above factors on treatment outcomes.

  • Reference #6 to the number of genotypes (6) in the second paragraph is too old and the number of HCV genotypes is currently 8.

Thank you for your comment. The information and the reference have been updated.

  • There is some redundancy in the methodology section and the supplementary data

Thank you, we have revised these sections

Minor comments:  

  • “95%CI” Is usually joined although there should be a space “95% CI” ->

Thank you for your input. The suggested change has been applied.

  • Line 51: (SVR) should be placed after “sustained virological response”

Thank you for your comment. We have changed it accordingly.

  • Line 77: “and their partners” should be deleted

Thank you for your comment. The suggested change has been applied.

  • Line 150: “Will compare” should be in the past

Thank you for your comments. The suggested change has been applied.

  • Some acronyms e.g., LFK, Doi,...etc. Should be spelled out and abbreviated at their first appearance and used as abbreviations later.

Thank you for your comment. We have ensured that acronyms are spelled out at their first appearance. However, in this case, "Doi" refers to a full word (the name of the author) and is not an abbreviation.

  • Figure 1 legend must be detailed so that the figure stands alone and explain what the stars in the figure mean and explain the mentioned abbreviations.

Thank you for your comment. The figure legend has been expanded to provide more details, including explanations for the abbreviations. The two asterisks (**) in figure (1) were left by mistake from the original version and do not have a specific meaning in the current context. We have corrected and updated the figure.

  • Line 308 needs a reference after “one study”. The same applies for line 322 after “Previous meta-analyses".

Thank you for your comment. References have been added.

Reviewer 2 Report

Comments and Suggestions for Authors

The references could be updated with more recent ones.

Author Response

  • The references could be updated with more recent ones.

Thank you for your comment. The references have been updated.

Reviewer 3 Report

Comments and Suggestions for Authors

Thanks for the chance to review this high quality systematic review and meta-analysis comparing sofosbuvir based regimens with and without ribavirin in the re-treatment of non-genotype 1 HCV infection. The study methodology and conclusions are clear and well laid out.

A few suggestions for improvement follow:

  1. Change the title to make clear that non-sofosbuvir based treatments were excluded from the metanalysis ie the conclusion is only regarding adding ribavirin to sofosbuvir-based regimens in re-treatment of non-G1 patients.
  2. The paper reads well, but the abstract is a little stilted and the English could be improved. eg line 28 change "different HCV genotypes" to "any non-1 HCV genotype"   line 29 "The additional ribavirin showed increased odds of developing adverse events" change to "The addition of ribavirin was associated with increased adverse effects"
  3. Overall I am happy that the paper is ready for publication with only minor revisions. 
Comments on the Quality of English Language

See above

Author Response

A few suggestions for improvement follow:

  1. Change the title to make clear that non-sofosbuvir based treatments were excluded from the metanalysis ie the conclusion is only regarding adding ribavirin to sofosbuvir-based regimens in re-treatment of non-G1 patients.

Thank you for your comment, we agree with the suggestion and have applied it accordingly.

  1. The paper reads well, but the abstract is a little stilted and the English could be improved. eg line 28 change "different HCV genotypes" to "any non-1 HCV genotype" line 29 "The additional ribavirin showed increased odds of developing adverse events" change to "The addition of ribavirin was associated with increased adverse effects"

Thank you for your suggestions. We have now made these changes.

  1. Overall I am happy that the paper is ready for publication with only minor revisions.

Thank you so much for providing us with feedback.

Reviewer 4 Report

Comments and Suggestions for Authors

The article by Hamran et al., is entitled “Efficacy and safety of adding ribavirin to direct-acting antivirals (DAAs) in re-treating non-genotype 1 hepatitis C- a systematic review and meta-analysis”.  Eight RCTs that contrasted sofosbuvir-based combinations with and without ribavirin were included in this review. The authors conclude that ribavirin did not assist in achieving sustained virological response (SVR) with a middle level certainty GRADE evidence as compared to sofosbuvir alone. Also, patients with different HCV genotypes did not benefit from using ribavirin in addition to sofosbuvir, according to subgroup analysis. There was a higher chance of adverse events and treatment termination when using added ribavirin. The manuscript is well-written and well-organized with minor editorial issues. However, although the data presented are of interest to scientists in the field, the review needs to focus not only on the methodology of their analyses and its validity but also on the detail of the patient populations examined in the different clinical trials reported as it has a direct connection to the outcome of these studies. The following concerns must be addressed by the authors before consideration for publication.

Major concerns:

  1. Since it directly affects the findings of these studies, the review must focus on the characteristics of the specific patient populations analyzed in the 8 clinical trials presented, in addition to the analysis's methodology and validity.
  2. Reference #6 to the number of genotypes (6) in the second paragraph is too old and the number of HCV genotypes is currently 8.
  3. There is some redundancy in the methodology section and the supplementary data.

Minor comments:

  1. “95%CI” is usually joined although there should be a space -->  95%CI
  2. Line 51: (SVR) should be placed after “sustained virological response”
  3. Line 77: “and their partners” should be deleted.
  4. Line 150: “will compare” should be in the past.
  5. Some acronyms e.g., Doi, LFK,…etc.  should be spelled out and abbreviated at their first appearance and used as abbreviated later.
  6. Figure 1 legend must be detailed so that the figure stand alone and explain what the stars in the figure mean and explain the mentioned abbreviations.
  7. Line 308 needs a reference after “one study”. The same applies for line 322 after “Previous meta-analyses”.

Author Response

  • Since it directly affects the findings of these studies, the review must focus on the characteristics of the specific patient populations analyzed in the eight clinical trials presented, in addition to the analysis's methodology and validity.
    Many thanks for your informative remarks on our manuscript. We appreciate being given the chance to improve our work in accordance with your input. We have a separate section in our paper that describes the patient populations studied across all of the eight included studies. We present a table that summarizes the characteristics of the studies. In addition, we have updated our discussion to reflect the possible impact of the above factors on treatment outcomes.

  • Reference #6 to the number of genotypes (6) in the second paragraph is too old and the number of HCV genotypes is currently 8.

Thank you for your comment. The information and the reference have been updated.

  • There is some redundancy in the methodology section and the supplementary data

Thank you, we have revised these sections

Minor comments:  

  • “95%CI” Is usually joined although there should be a space “95% CI” ->

Thank you for your input. The suggested change has been applied.

  • Line 51: (SVR) should be placed after “sustained virological response”

Thank you for your comment. We have changed it accordingly.

  • Line 77: “and their partners” should be deleted

Thank you for your comment. The suggested change has been applied.

  • Line 150: “Will compare” should be in the past

Thank you for your comments. The suggested change has been applied.

  • Some acronyms e.g., LFK, Doi,...etc. Should be spelled out and abbreviated at their first appearance and used as abbreviations later.

Thank you for your comment. We have ensured that acronyms are spelled out at their first appearance. However, in this case, "Doi" refers to a full word (the name of the author) and is not an abbreviation.

  • Figure 1 legend must be detailed so that the figure stands alone and explain what the stars in the figure mean and explain the mentioned abbreviations.

Thank you for your comment. The figure legend has been expanded to provide more details, including explanations for the abbreviations. The two asterisks (**) in figure (1) were left by mistake from the original version and do not have a specific meaning in the current context. We have corrected and updated the figure.

  • Line 308 needs a reference after “one study”. The same applies for line 322 after “Previous meta-analyses".

Thank you for your comment. References have been added.